# Evaluation of an on-site sanitation intervention against childhood diarrhea and acute respiratory infection 1 to 3.5 years after implementation: Extended follow-up of a cluster-randomized controlled trial in rural Bangladesh

Jesse D. Contreras[1], Mahfuza Islam[2], Andrew Mertens[3], Amy J. Pickering[4], Benjamin F. Arnold[5], Jade Benjamin-Chung[6], Alan E. Hubbard[3], Mahbubur Rahman[2], Leanne Unicomb[2], Stephen P. Luby[7], John M. Colford, Jr[3], Ayse Ercumen[1] *

1 Department of Forestry and Environmental Resources, North Carolina State University, Raleigh, North Carolina, United States of America, 2 Environmental Interventions Unit, Infectious Disease Division, icddr,b, Dhaka, Bangladesh, 3 Division of Epidemiology and Biostatistics, School of Public Health, University of California, Berkeley, California, United States of America, 4 Department of Civil and Environmental Engineering, University of California, Berkeley, California, United States of America, 5 Francis I. Proctor Foundation, University of California, San Francisco, California, United States of America, 6 Department of Epidemiology and Population Health, Stanford University, Palo Alto, California, United States of America, 7 Infectious Diseases and Geographic Medicine, Stanford University, Stanford, California, United States of America

* aercume@ncsu.edu

## Abstract

### Background

Diarrhea and acute respiratory infection (ARI) are leading causes of death in children. The WASH Benefits Bangladesh trial implemented a multicomponent sanitation intervention that led to a 39% reduction in the prevalence of diarrhea among children and a 25% reduction for ARI, measured 1 to 2 years after intervention implementation. We measured longer-term intervention effects on these outcomes between 1 to 3.5 years after intervention implementation, including periods with differing intensity of behavioral promotion.

### Methods and findings

WASH Benefits Bangladesh was a cluster-randomized controlled trial of water, sanitation, hygiene, and nutrition interventions (NCT01590095). The sanitation intervention included provision of or upgrades to improved latrines, sani-scoops for feces removal, children's potties, and in-person behavioral promotion. Promotion was intensive up to 2 years after intervention initiation, decreased in intensity between years 2 to 3, and stopped after 3 years. Access to and reported use of latrines was high in both arms, and latrine quality was significantly improved by the intervention, while use of child feces management tools was low. We

**Data Availability Statement:** The trial protocol, pre-registered analysis plan, deidentified participant data, and analysis scripts are freely available at OSF (https://osf.io/6u7cn/).

**Funding:** This study was supported by Grant R01HD078912 from the National Institutes of Health and in part by Grant 0PPGD759 from the Bill and Melinda Gates Foundation to the University of California, Berkeley (authors BFA, SFL, JMC, and AE). The funders approved the study design, but had no role in data collection, data analysis, or manuscript preparation.

**Competing interests:** The authors have declared that no competing interests exist.

enrolled a random subset of households from the sanitation and control arms into a longitudinal substudy, which measured child health with quarterly visits between 1 to 3.5 years after intervention implementation. The study period therefore included approximately 1 year of high-intensity promotion, 1 year of low-intensity promotion, and 6 months with no promotion. We assessed intervention effects on diarrhea and ARI prevalence among children <5 years through intention-to-treat analysis using generalized linear models with robust standard errors. Masking was not possible during data collection, but data analysis was masked. We enrolled 720 households (360 per arm) from the parent trial and made 9,800 child observations between June 2014 and December 2016. Over the entire study period, diarrheal prevalence was lower among children in the sanitation arm (11.9%) compared to the control arm (14.5%) (prevalence ratio [PR] = 0.81, 95% CI 0.66, 1.00, $p = 0.05$; prevalence difference [PD] = −0.027, 95% CI −0.053, 0, $p = 0.05$). ARI prevalence did not differ between sanitation (21.3%) and control (22.7%) arms (PR = 0.93, 95% CI 0.82, 1.05, $p = 0.23$; PD = −0.016, 95% CI −0.043, 0.010, $p = 0.23$). There were no significant differences in intervention effects between periods with high-intensity versus low-intensity/no promotion. Study limitations include use of caregiver-reported symptoms to define health outcomes and limited data collected after promotion ceased.

## Conclusions

The observed effect of the WASH Benefits Bangladesh sanitation intervention on diarrhea in children appeared to be sustained for at least 3.5 years after implementation, including 1.5 years after heavy promotion ceased. Existing latrine access was high in the study setting, suggesting that improving on-site latrine quality can deliver health benefits when latrine use practices are in place. Further work is needed to understand how latrine adoption can be achieved and sustained in settings with low existing access and how sanitation programs can adopt transformative approaches of excreta management, including safe disposal of child and animal feces, to generate a hygienic home environment.

## Trial registration

ClinicalTrials.gov; NCT01590095; https://clinicaltrials.gov/ct2/show/NCT01590095.

Author summary

**Why was this study done?**

- Although sanitation is believed to be crucial for preventing diarrheal disease in children, most randomized trials assessing household sanitation interventions have found no effect on diarrheal disease.

- The sustainability of health effects among effective sanitation interventions is unknown, as almost all trials have measured outcomes between 1 to 2 years after implementation.

- The roles of behavioral promotion and infrastructure improvements in sanitation trials have not been differentiated and may explain varying intervention effects across contexts.

## What did the researchers do and find?

- We assessed the effects of an on-site sanitation intervention, comprising latrine upgrades, child feces management tools, and behavioral promotion, on childhood diarrheal disease and acute respiratory infection (ARI) between 1 to 3.5 years after intervention implementation.

- The prevalence of diarrheal disease among children under 5 years was significantly reduced among intervention participants, but there was no effect on ARI.

- Intervention effects did not wane over time, including after high-intensity promotion efforts were replaced by low-intensity/no behavioral promotion.

## What do these findings mean?

- The intervention's observed effects on diarrheal disease persisted over time, including 1.5 years after behavioral promotion was tapered and ceased.

- Baseline latrine access and use was high in the study setting but the intervention significantly improved latrine quality. In a setting with high background latrine use, improvements to latrine quality can effectively reduce diarrheal disease.

## Introduction

Diarrheal disease and lower respiratory tract infections are among the leading causes of death for children under 5 years worldwide. Observational evidence suggests diarrheal disease can lead to increased risk of subsequent respiratory infections in children [1,2]. Thus, interventions to prevent diarrheal disease may also prevent acute respiratory infections (ARIs). Access to adequate sanitation has long been viewed as an important tool for preventing diarrheal disease and substantial observational evidence linked the 2 during the 20th century [3]. Systematic meta-analyses of experimental and matched observational data have estimated that access to improved sanitation reduces the risk of diarrheal disease in children under 5 years by about 25% on average [4,5]. However, further subgroup analysis in a recent meta-analysis shows that those average effects are primarily driven by studies on expanded sewerage access and interventions that improved water quality alongside sanitation, while solely latrine-based intervention trials have found no effect on diarrheal disease on average [6].

Only 2 out of 6 latrine-based controlled trials identified in the most recent meta-analysis found an impact on diarrheal disease and only one of these reduced diarrheal disease specifically among children [4]. Both effective interventions were large scale, internationally funded projects with characteristics that restrict external validity. In Honduras, sanitation was a single component of a nationwide development project [7]. Communities were offered a menu of project options to choose from, including sanitation, and the selected project was delivered at the community level. However, the estimated effect of sanitation on diarrhea in that study could be explained by confounding due to urban–rural differences between intervention and control communities [7]. In Mozambique, a community-led sanitation intervention reduced undefined "water-related diseases" measured with a 6-month recall period among all ages but

had no impact on the same outcome among children under 5 years or children under 3 years [8,9]. The 5 trials of latrine-based interventions that did not affect childhood diarrheal disease all depended on promotion to encourage construction of latrines in areas with low baseline latrine access; access to latrines remained relatively low in intervention areas, which possibly contributed to the null effects [8,10–13]. A systematic review of sanitation interventions found that interventions to promote latrine construction only modestly impact latrine ownership on average [14]. Additionally, access to a latrine does not equate to latrine use. Promotion of latrine use can achieve behavior change in some contexts, although behavioral interventions have reported mixed success in achieving latrine use among intervention recipients [14,15]. Also, most studies have focused on documenting short-term uptake, while behavior change has not been sustained in a small number of studies that assessed longer-term latrine use [15].

More recently, 3 controlled trials estimated the efficacy of direct provision of high-quality on-site sanitation facilities on child health [16–18]. The Maputo Sanitation (MapSan) controlled before-and-after study provided pour-flush toilets and septic tanks to household clusters in urban Mozambique and found no change in child diarrhea [18]. The WASH Benefits Kenya and Bangladesh trials provided latrine upgrades, child feces management tools, and behavioral promotion in rural Kenya and Bangladesh. The Kenya trial found no effect on diarrheal disease [17], soil-transmitted helminth infections [19], and ARI [20] from the sanitation intervention. In Bangladesh, the sanitation intervention led to prevalence reductions of 39% for diarrheal disease, 25% for *Giardia* infections, 29% for *Trichuris trichiura* infections, and 25% for ARI [16,21–23].

Notably, almost all latrine-based intervention trials conducted to date evaluated child health outcomes approximately 1 to 2 years after the intervention was implemented [9,11–13,16,17]. In 1 trial in Tanzania, outcomes for early recipients of the intervention were measured up to 3 years after implementation; the timing of outcome measurement ranged between 1 to 2 years for other participants [10]. However, that trial considered the entire data collection period as a single follow-up round and did not assess variation in intervention effects on child health over time. The long-term sustainability of water, sanitation, and hygiene interventions was recently recognized as a significant gap in the existing evidence [24]. Interventions that depend on individual behavior change may see reduced effectiveness over time as recipients discontinue adherence, as evidenced by a decrease in the average effectiveness of household water treatment interventions over time [25]. Because few sanitation trials to date have demonstrated health impacts and have only assessed effects over relatively short follow-up periods, little is known about how the effect of sanitation interventions on health might change over time.

In this study, we assessed the longer-term effects of the WASH Benefits Bangladesh sanitation intervention, which demonstrated reduced diarrheal disease and ARI among children 1 to 2 years after intervention implementation. We used longitudinal data from a random subset of trial participants, collected between 1 to 3.5 years after intervention implementation, including periods with different intensity of behavioral promotion.

## Methods

### Randomization and masking

The WASH Benefits trial was a cluster-randomized controlled trial that implemented water, sanitation, hygiene, and nutrition interventions in rural Bangladesh and Kenya (NCT01590095). The trial enrolled households of pregnant women in their first or second trimester. In Bangladesh, enrolled households were grouped into clusters of 6 to 8 spatially adjacent households. Clusters were spaced at least 15 minutes walking distance apart to reduce potential spillover effects. Eight adjacent clusters formed a study block. Using a random

number generator, each cluster within a block was randomly assigned to 1 of 6 intervention arms (water; sanitation; handwashing; water, sanitation, and handwashing (WASH); nutrition; or WASH and nutrition) or into a double-sized control arm, creating geographically matched clusters within each block. The interventions comprised visible materials and in-person visits, so neither participants nor field staff were masked to intervention assignment. Data analysis was masked by replacing true intervention assignment with a re-randomized assignment variable.

## Interventions

The sanitation intervention included double-pit pour flush improved latrines, a sani-scoop for the removal of child and animal feces, and a children's potty, along with in-person behavioral promotion on product use and maintenance. The intervention was tailored to local preferences through extensive piloting [26]. Households in the study area were most commonly part of multifamily compounds. The intervention was delivered at the compound level with emphasis on the index household, where the pregnant enrollee lived. New latrines were constructed to replace any latrine in the compound that did not have a slab or water seal, or that failed to prevent surface runoff of feces. A new latrine was provided for the index household if they did not own a latrine. At baseline, 46% of index households did not own a latrine; households without their own latrine in this setting typically share a latrine within the family compound. Sani-scoops were provided to all households in the compound, and potties were provided to all households that had children <3 years. While promotion activities primarily targeted the index household, other households in the compound were invited to participate and participated at will. Promoters did not visit households in the control arm. Behavioral promotion in the intervention arm was intensive (6 household visits per month on average by community health promoters) for the first 2 years after intervention initiation until data collection for the parent trial was completed. Promotion intensity gradually decreased between years 2 to 3, beginning with 1 visit per month, and all promotion activities stopped approximately 3 years after intervention initiation. Spot check observations and structured questionnaires demonstrated that the intervention increased access to and use of hygienic latrines throughout our study period (1 to 3.5 years after implementation), while uptake was low and inconsistent for sani-scoops and potties [27].

## Substudy design

A random subset of index households from the sanitation and control arms of the parent trial were enrolled into a longitudinal substudy that measured environmental contamination. In each block, we enrolled 4 households from the sanitation cluster and 4 households from 1 of 2 control clusters, which maintained the geographic matching from the parent trial. Trained field staff visited enrolled households approximately every 4 months for 2.5 years, for a total of 8 visits between 1 to 3.5 years after intervention implementation. The data collection period therefore included approximately 1 year of high-intensity promotion, 1 year of low-intensity promotion, and 6 months with no promotion. At each visit, field staff collected samples from the household environment and administered a structured survey on children's health, including diarrheal symptoms, respiratory symptoms, and bruising/abrasion, using the same survey instrument as the parent trial. Symptoms were reported for each child under 5 years present in the compound at the time of the survey by the child's primary caregiver. We followed CONSORT guidelines in reporting the design and results of this trial (S1 CONSORT Checklist).

## Ethics

Written informed consent was provided by the primary caregiver of enrolled children in the local language (Bengali). Human subjects committees at the International Centre for Diarrhoeal Disease Research, Bangladesh (icddr,b) (PR-11063), University of California, Berkeley (2011−09−3652), and Stanford University (25863) approved the study protocol.

## Study outcomes

We analyzed data according to a preregistered analysis plan (available at: https://osf.io/r4kbp/). Our primary outcomes were caregiver-reported diarrheal disease and ARI in the past 7 days among children under 5 years. We defined diarrhea as at least 3 loose or watery stools within 24 hours or at least 1 stool with blood, consistent with the definition used in the parent trial [16]. We defined ARI as persistent cough, panting, wheezing, or difficulty breathing [23]. We included bruising/abrasion within the past 7 days as a negative control outcome to detect potential bias. There is no plausible relationship between the intervention and bruising/abrasion, so any apparent effects on bruising/abrasion could indicate bias in our primary analyses.

## Statistical analysis

We estimated the prevalence of diarrheal disease and ARI during each month of follow-up separately for the sanitation and control groups to observe temporal and seasonal changes. We estimated Bayesian credible intervals to account for sampling error, which used a beta distribution to restrict proportion estimates to values greater than 0. Initial inspection of data from the control arm revealed that the prevalence of child health outcomes was higher in this substudy compared to the parent trial. As the substudy was conducted among a random subset of the parent trial participants, and the data collection periods did not fully overlap between the studies, sampling error, temporal variation in disease risk, and differences in children's ages could lead to differing disease prevalence between the studies. To investigate sampling error, we calculated the prevalence of diarrheal disease and ARI in the parent trial dataset among the subset of compounds that participated in the substudy. To investigate temporal variability and the impact of child age, we compared the monthly prevalence of diarrheal disease during overlapping periods when both studies were active using credible intervals to account for sampling error.

To measure the effect of the intervention, we pooled child observations across all 8 sampling rounds. We estimated a prevalence ratio (PR) and prevalence difference (PD) comparing children living in sanitation and control households for each outcome using intention-to-treat analysis through generalized linear models. Robust sandwich standard errors were estimated using study block as the unit of clustering to account for cluster randomization and repeated observations of individual children. Models were adjusted for block to control for geographical matching. Per our pre-analysis plan, we did not adjust for additional covariates because randomization led to good balance of baseline covariates between arms among the subset of households enrolled in the substudy [28]. The sample size of this substudy was based on power calculations for a separate analysis of fecal contamination. Before this analysis, we estimated minimum detectable effects for diarrheal disease and ARI, given the existing sample size and the observed prevalence values in the control arm of the substudy. The available observations yielded 80% power to detect a PR of 0.77 and a PD of −3.3 percentage points for diarrheal disease and a PR of 0.84 and a PD of −3.5 percentage points for ARI.

## Effect modification

We assessed potential effect modification by living in index versus non-index households in the compound, periods of high-intensity promotion versus low-intensity or no promotion, and monsoon versus dry season. Index households were the primary targets of the intervention, both in terms of provision of hardware and behavior change promotion. Due to the rolling nature of the WASH Benefits trial, promotion intensity was not tapered or ceased on specific dates. Instead, promotion intensity was tapered for households around the time they completed the final visit of the parent trial. Therefore, we operationalized promotional intensity at the household level by comparing the date of each substudy observation to the household's final survey date for the parent trial. Observations during high-intensity promotion were those that preceded the parent trial's final survey, and observations during low-intensity/ no promotion were those that followed the parent trial's final survey. We defined the monsoon season using daily rainfall data recorded by the Bangladesh Meteorological Department at 3 weather stations closest to the study region for the years 2014 to 2016 [29]. We calculated 5-day rolling averages of daily rainfall at each station and defined the start of the monsoon season for each year as the first day with a 5-day rolling average rainfall of 10 mm or greater at any station; we defined the end of the monsoon season for each year as the last day with a 5-day rolling average rainfall of 10 mm or greater at any station. Under this definition, the monsoon seasons were April 2 to September 27, 2014; March 31 to September 25, 2015; and March 30 to October 30, 2016. We assessed effect modification by including an interaction term between each variable and study arm in regression models for each outcome.

## Results

We enrolled 720 index households and their familial compounds, divided evenly between the sanitation and control arms (Fig 1). Eighty percent of households provided data for all 8 survey rounds and 96% completed at least 6 surveys. Loss to follow-up was similar between arms, and participants that left versus completed the study were similar in their characteristics [28]. At the first visit, approximately 1 year after the intervention was implemented, 10 compounds in the control arm (3%) had no latrine, while all compounds in the sanitation arm had at least 1 latrine. During the same visit, 100% of index households in the sanitation arm reported primarily using a latrine in their compound for defecation, compared to 97% of controls, and 86% of sanitation index households and 77% of controls reported that adults always used a latrine for defecation. Latrines in the sanitation arm were less likely to be shared with other households than among controls (20% versus 54%), and there were an average of 4.2 compound residents per latrine in the sanitation arm compared to 7.1 in the control arm. Eighty-two percent of index households in the sanitation arm reported that their primary latrine was built or upgraded through the intervention. Compared to the primary latrines used by index households in the control arm, those in the sanitation arm were more likely to be pour flush (99% versus 76%) and to have a functional water seal (95% versus 46%). Over 95% of primary latrines in both arms had slabs.

In total, we enrolled 1,789 children under 5 years (922 in the sanitation arm and 867 in the control arm) in the substudy, for a total of 9,800 child observations (5,088 in the sanitation arm and 4,712 in the control arm) over 8 data collection rounds between June 2014 and December 2016 (S1 Fig). The rate of new births in the compound was similar between arms, with new births reported at 9% and 10% of visits in the sanitation and control arms, respectively. The median age of children at the first and last visits was approximately 15 and 38 months, respectively, and observations were divided evenly between male and female children. Child age and sex were well balanced between the intervention and control arms.

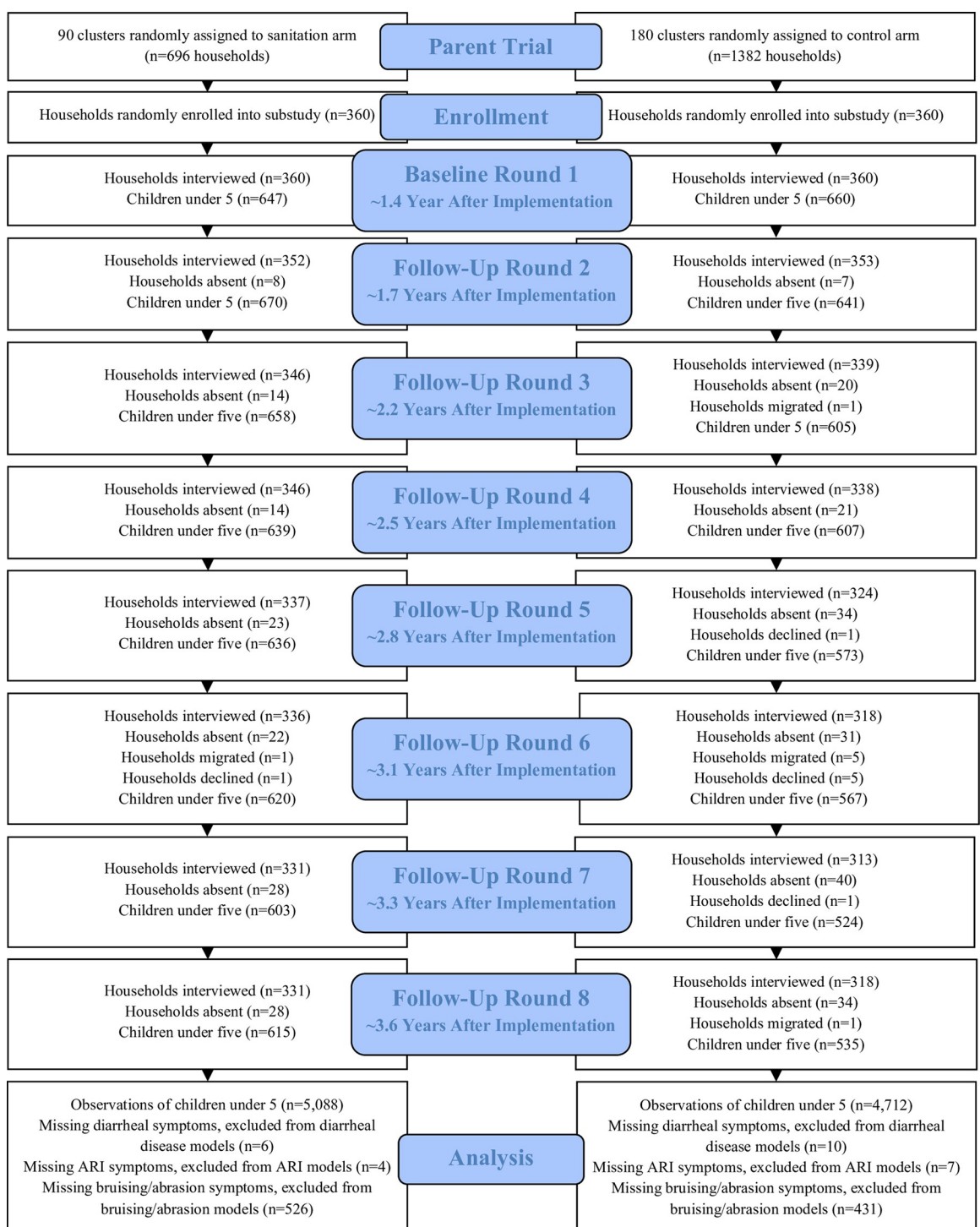

**Fig 1.** Flow chart of study participants in sanitation (left) and control (right) arms over each survey round. Survey rounds labeled with the median approximate time passed since intervention implementation.

The overall 7-day prevalence of diarrheal disease among children under 5 years was 13.2% (Fig 2). The prevalence of diarrheal disease was highest during the first survey round (16.5%) and lowest during the fourth round (10.9%). The overall 7-day prevalence of ARI was 22.0% (Fig 3); the prevalence was highest during the second survey round (30.7%) and lowest during the last round (14.1%). The overall prevalence of bruising/abrasion was 8.2%; the prevalence was highest during the sixth round (10.6%) and lowest during the first round (5.7%). Fewer than 1% of observations were missing health outcome data and were not included in this analysis. Prevalence estimates for all outcomes were higher in this substudy compared to the parent trial, including during overlapping data collection periods and after limiting parent trial data to substudy participants (S2–S4 Figs).

**Fig 2. Prevalence of diarrheal disease among children under 5 years by intervention arm and month of follow-up.** Shaded bands represent 95% credible intervals around each prevalence estimate. Monsoon and dry seasons are indicated with vertical dashed lines.

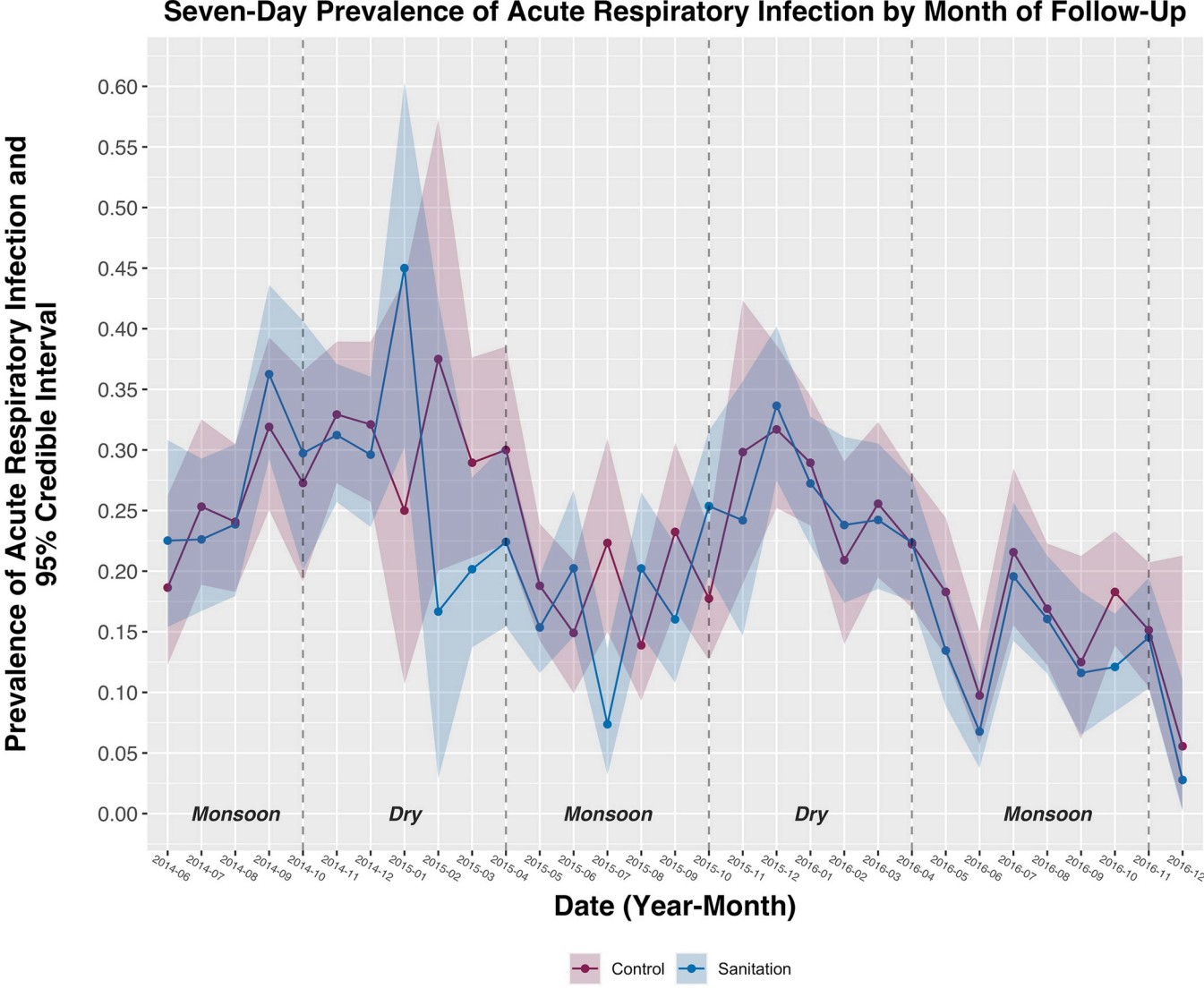

**Fig 3. Prevalence of ARI among children under 5 years by intervention arm and month of follow-up.** Shaded bands represent 95% credible intervals around each prevalence estimate. Monsoon and dry seasons are indicated with vertical dashed lines. ARI, acute respiratory infection.

### Intervention effects

The prevalence of diarrheal disease among children under 5 years was 19% lower in the sanitation arm compared to the control arm [Table 1; PR = 0.81 (95% CI 0.66, 1.00, $p = 0.05$); PD = −0.027 (95% CI −0.053, 0, $p = 0.05$)]. ARI prevalence did not significantly differ between arms [Table 1; PR = 0.93 (95% CI 0.82, 1.05, $p = 0.23$); PD = −0.016 (95% CI −0.043, 0.010, $p = 0.23$)]. The prevalence of bruising/abrasion was 13% lower in the sanitation arm compared to controls, but the confidence intervals included the null [Table 1; PR = 0.87 (95% CI 0.73, 1.02, $p = 0.09$); PD = −0.012 (95% CI −0.025, 0.002, $p = 0.09$)].

### Effect modification

Intervention effects on diarrheal disease were modified between children living in the index household (i.e., primary intervention recipients in the compound) versus other households in

**Table 1. Effects of the sanitation intervention on primary child health outcomes and negative control outcome.**

| | Prevalence in sanitation arm | Prevalence in control arm | PR (95% CI) | PD (95% CI) |
|---|---|---|---|---|
| *Primary child health outcomes* | | | | |
| Diarrheal disease | 11.9% | 14.5% | 0.81 (0.66, 1.00) | −0.027 (−0.053, 0.000) |
| Acute respiratory infection | 21.3% | 22.7% | 0.93 (0.82, 1.05) | −0.016 (−0.043, 0.010) |
| *Negative control outcome* | | | | |
| Bruising/abrasion | 7.6% | 8.8% | 0.87 (0.73, 1.02) | −0.012 (−0.025, 0.002) |

All outcomes reported using a 7-day recall period.

PD, prevalence difference; PR, prevalence ratio.

the compound ($p = 0.15$) and season ($p = 0.02$), but not by differing promotion intensity over time (i.e., high-intensity promotion in years 1 to 2 after intervention implementation versus low-intensity/no promotion in years 2 to 3.5) ($p = 0.86$) (Table 2). Among children living in index households, the prevalence of diarrheal disease in the sanitation arm (10.8%) was 26% lower than in the control arm (14.5%) [Table 2; PR = 0.74 (95% CI 0.61, 0.91, $p < 0.01$); PD = −0.037 (95% CI −0.062, −0.011, $p < 0.01$)]. The intervention had no effect on diarrheal disease among children living in other households within the compound. During monsoon seasons, the prevalence of diarrheal disease in the sanitation arm (11.3%) was 27% lower than in the control arm (15.3%) [Table 2; PR = 0.73 (95% CI 0.57, 0.92, $p < 0.01$); PD = −0.041 (95% CI −0.072, −0.011, $p < 0.01$)]. The intervention had no effect on diarrheal disease during dry seasons. There was no evidence of effect modification by any variable for ARI or bruising/abrasion (Table 2).

## Discussion

We found that the WASH Benefits Bangladesh sanitation intervention was associated with a reduction in the prevalence of diarrheal disease among children under 5 years from 14.5% to 11.9% (PR = 0.81, 95% CI 0.66, 1.00, $p = 0.05$) between 1 to 3.5 years after the intervention was implemented, but the intervention had no effect on ARI. Our findings indicate that the previously reported reduction in diarrheal disease, measured at 1 and 2 years after intervention implementation, was sustained over approximately 1.5 additional years [16]. Intervention effects persisted as promotional activities were tapered from high-frequency visits during the first year of this substudy to low-frequency visits during the second year and no visits during the final 6 months of data collection.

The WASH Benefits Bangladesh intervention remains 1 of only 2 controlled latrine-based interventions that has successfully reduced childhood diarrheal disease [7]. Intensive promotional activities delivered directly to intervention recipients may have enabled these rare effects. Promotional activities were targeted to index households, although all households in the compound were invited to participate. Intervention effects were found exclusively among index households; the sanitation intervention reduced diarrhea by 26% among children living in index households, while there was no effect for children living in other households in the compound. This finding further supports the important role of intense promotion. However, our effect modification results for periods with different promotion intensity suggest that frequent promotion does not need to be continued indefinitely and health effects may be sustained after tapering and possibly ceasing behavioral promotion if adequate sanitation habits are built. We also note that at the baseline of the parent trial, roughly half of index households in the sanitation and control arms owned their own latrine, while those without their own latrine typically shared with another household in the compound [16]. Index households were

**Table 2. Effect modification by household of residence (children living in index household[a] vs. in other households within compound), periods of high-intensity vs. low-intensity or no promotion (during vs. after parent trial completion), and season (monsoon vs. dry).**

| | Prevalence (sanitation arm) | Prevalence (control arm) | PR (95% CI) | p-Value for interaction on multiplicative scale | PD (95% CI) | p-Value for interaction on additive scale |
|---|---|---|---|---|---|---|
| **Diarrheal disease** | | | | | | |
| *Household* | | | | 0.11 | | 0.15 |
| Index household | 10.8% | 14.5% | 0.74 (0.61, 0.91) | – | −0.037 (−0.062, −0.011) | – |
| Other households in compound | 14.0% | 14.5% | 0.94 (0.69, 1.29) | – | −0.007 (−0.050, 0.037) | – |
| *Promotion intensity* | | | | 0.85 | | 0.86 |
| High (during parent trial) | 13.3% | 16.0% | 0.83 (0.66, 1.04) | – | −0.028 (−0.619, 0.006) | – |
| Low or none (after parent trial) | 11.3% | 13.8% | 0.81 (0.64, 1.02) | – | −0.025 (−0.054, 0.003) | – |
| *Season* | | | | 0.02 | | 0.02 |
| Monsoon season | 11.3% | 15.3% | 0.73 (0.57, 0.92) | – | −0.041 (−0.072, −0.011) | – |
| Dry season | 12.9% | 13.3% | 0.96 (0.76, 1.22) | – | −0.005 (−0.035, 0.026) | – |
| **ARI** | | | | | | |
| *Household* | | | | 0.87 | | 0.89 |
| Index household | 21.2% | 22.9% | 0.93 (0.81, 1.06) | – | −0.017 (−0.046, 0.012) | – |
| Other households in compound | 21.4% | 22.2% | 0.94 (0.78, 1.13) | – | −0.014 (−0.054, 0.026) | – |
| *Promotion intensity* | | | | 0.53 | | 0.70 |
| High (during parent trial) | 25.8% | 26.5% | 0.97 (0.83, 1.13) | – | −0.010 (−0.050, 0.031) | – |
| Low or none (after parent trial) | 19.2% | 20.8% | 0.91 (0.78, 1.06) | – | −0.018 (−0.049, 0.012) | – |
| *Season* | | | | 0.40 | | 0.50 |
| Monsoon season | 18.2% | 20.1% | 0.89 (0.77, 1.04) | – | −0.022 (−0.050, 0.006) | – |
| Dry season | 25.8% | 26.5% | 0.96 (0.83, 1.11) | – | −0.009 (−0.047, 0.030) | – |
| **Bruising/abrasion** | | | | | | |
| *Household* | | | | 0.83 | | 0.93 |
| Index household | 8.0% | 9.2% | 0.88 (0.72, 1.08) | – | −0.011 (−0.028, 0.007) | – |
| Other households in compound | 6.8% | 8.0% | 0.85 (0.63, 1.14) | – | −0.012 (−0.034, 0.010) | – |
| *Promotion intensity* | | | | 0.54 | | 0.34 |
| High (during parent trial) | 6.1% | 6.7% | 0.92 (0.73, 1.17) | – | −0.005 (−0.020, 0.011) | – |
| Low or none (after parent trial) | 8.3% | 9.9% | 0.84 (0.69, 1.03) | – | −0.016 (−0.034, 0.003) | – |
| *Season* | | | | 0.40 | | 0.38 |
| Monsoon season | 7.8% | 8.6% | 0.92 (0.73, 1.15) | – | −0.007 (−0.025, 0.011) | – |

(*Continued*)

**Table 2.** (Continued)

|  | Prevalence (sanitation arm) | Prevalence (control arm) | PR (95% CI) | *p*-Value for interaction on multiplicative scale | PD (95% CI) | *p*-Value for interaction on additive scale |
|---|---|---|---|---|---|---|
| Dry season | 7.3% | 9.2% | 0.80 (0.63, 1.01) | – | −0.019 (−0.038, 0.000) | – |

[a]Index household refers to household where the pregnant enrollee lived; index households were the primary intervention targets.

All outcomes reported using a 7-day recall period.

ARI, acute respiratory infection; PD, prevalence difference; PR, prevalence ratio.

the only households guaranteed a new improved latrine if they did not have one, while other latrines in the compound were eligible to be upgraded if necessary. Thus, all index households in the sanitation arm had their own latrine after the intervention was implemented, and increased personal access to an improved latrine may have led to the unique effects among index household children.

Notably, WASH Benefits Bangladesh was conducted in a setting with high baseline access to on-site latrines. At the onset of this substudy, approximately 1 year after intervention implementation in the sanitation arm, 97% of index households in the control arm reported using a latrine within their compound (versus 100% of index households in the sanitation arm) and 77% reported that adults used the latrine exclusively for defecation (versus 86% in the sanitation arm). As the control arm did not receive any intervention or behavior change promotion, we expect the conditions among controls to represent baseline conditions before the onset of the parent trial. Only 54% of index households in either arm owned their own latrine at the baseline of the parent trial [16]. Providing a new, personal latrine to any index household that did not have one may have helped increase latrine use in the sanitation arm by reducing barriers associated with intra-compound latrine sharing, such as privacy or availability concerns. However, high reported latrine use in the control arm indicates that latrine habits appear to have been in place in the community even without an intervention, and existing habits and norms may have facilitated sustained maintenance and use of latrines among intervention recipients beyond the period of intensive promotion.

In addition, the sanitation intervention included compound-wide improvements in latrine quality. This included latrine upgrades among index households that already owned a latrine at baseline and upgrades to any other existing latrines in the compound. Compared to controls, latrines primarily used by index households in the sanitation arm were more likely to be pour-flush (99% versus 76%), have a functional water seal (95% versus 46%), and less likely to be shared with other households (20% versus 54%). Our findings therefore indicate that, in a setting with existing latrine use habits and prevalent latrine sharing, provision of personal latrines and improvements to latrine quality achieved a sustained reduction in child diarrhea. Child potties and sani-scoops did not likely contribute to intervention effects, as use of these products was low and inconsistent, and child feces management practices remained poor in the sanitation arm [27]. We note that the intervention did not address communities' broader sanitation system, such as pit emptying practices or animal waste management. In addition, the parent WASH Benefits trial intervened in only around 10% of households within each community due to eligibility criteria. A broader sanitation approach may provide stronger effects on child health and measurably reduce levels of environmental fecal contamination, which this intervention did not achieve [28,30,31].

One limitation of this study was the use of caregiver-reported disease symptoms. Notably, the prevalence of both diarrheal disease and ARI was higher in this substudy compared to the parent trial. During the period when both studies were ongoing and the ages of children in

both studies were similar, the 7-day prevalence of diarrheal disease among controls was 16.0% in this substudy versus 5.5% in the parent trial, and the prevalence of ARI among controls was 26.5% in this substudy versus 8.8% in the parent trial. One possible cause of these differences is sampling error among substudy participants. However, we found that the differences in prevalence remained after limiting parent trial data to substudy participants, suggesting that the discrepancy is not due to sampling error. Another potential cause for the discrepancy is measurement error. We used identical survey questions to measure child health in both studies and field staff were trained by the same organization (icddr,b) using similar established procedures. However, the parent trial used a lengthier questionnaire and participants may have been less likely to report diarrheal cases due to survey fatigue. Furthermore, this substudy was primarily designed as an environmental assessment of fecal contamination, and most of the household visits were devoted to environmental sample collection. In contrast, data collection visits for the parent trial were more focused on the health questionnaire along with anthropometric measurements of children. It is possible that participants in the parent trial reported fewer cases of child health outcomes due to courtesy or social desirability due to the focus on health. If correct, these explanations would suggest that the prevalence measured in this substudy more closely approximates true population parameters. Previous studies conducted in the same area estimated the prevalence of diarrheal disease among children between 10% to 18%, further supporting the validity of prevalence estimates of this substudy, while the 5.7% prevalence measured in the parent trial was lower than anticipated [32–34].

Regardless of differences in absolute prevalence estimates, both studies found a reduction in diarrhea prevalence from the sanitation intervention. The effects were similar on the absolute scale (PD = −0.027, 95% CI: −0.053, 0 in this study and −0.022, 95% CI: −0.034, −0.010 in the parent trial) but differed on the relative scale (PR = 0.81, 95% CI: 0.66, 1.00 in this study and 0.61, 95% CI: 0.46, 0.81 in the parent trial). The magnitude and direction of bias in effect estimates from measurement error depends on the rate of over- or underreporting of symptoms and if misclassification was differential by study arm; when symptoms are underreported and underreporting is non-differential by arm, relative estimates of effect are unbiased while absolute effects are attenuated [35]. We cannot rule out differential measurement error in either study. In this substudy, we found a borderline reduction in the caregiver-reported prevalence of bruising and abrasion in the sanitation arm, suggesting potential underreporting of health symptoms by intervention recipients. However, there was no intervention effect on ARI, suggesting no overall underreporting across different health outcomes. Two additional pieces of evidence strongly suggest that the intervention did truly reduce the prevalence of diarrheal disease. First, the same intervention reduced the prevalence of *Giardia* infection by 25% [21] and *Trichuris trichiura* infection by 29% [22], measured by analysis of child stool samples, demonstrating a true impact on enteric pathogen transmission using objectively measured outcomes. Second, both the parent trial and this substudy found distinct seasonal effects of the intervention on diarrheal disease. If participants in the intervention arm underreported diarrheal symptoms due to courtesy or social desirability, we would expect the bias to be consistent across seasons. Instead, intervention effects were exclusively found during monsoon seasons when environmental fecal contamination is more widespread [36]. These seasonal effects are more credibly explained by modification of true intervention effects by season.

Overall, the results of this study suggest that the observed effect of the WASH Benefits Bangladesh sanitation intervention on diarrheal disease in children was sustained for at least 3.5 years after implementation, including 1 year after behavioral promotion was tapered and 6 months after promotion was ceased. While intensive early promotion in this efficacy trial may have enabled effects to persist beyond the duration of promotion, high existing access to and use of latrines indicates that sanitation norms may already have been in place, and health

benefits in this setting stemmed primarily from improvements in latrine quality. Further work is needed to understand how latrine adoption can be achieved and sustained in settings with low existing access to make larger strides along the sanitation ladder and how sanitation programs can move toward broader approaches of excreta management to effectively reduce fecal contamination in the environment and further improve health.

## Supporting information

**S1 CONSORT Checklist. CONSORT 2010 checklist of information to include when reporting a randomized trial.**
(PDF)

**S1 Fig. Number of individual observations of children under 5 years in the WASH Benefits Bangladesh parent trial and this substudy.** Parent trial data includes substudy participants and those not enrolled in the substudy. Observations are separated by approximate follow-up time after the sanitation intervention was implemented in participating households.
(PDF)

**S2 Fig. Comparison of the prevalence of diarrheal disease among children under 5 years by month of follow-up in the control arms of the parent trial and this substudy.** Prevalence estimates shown separately for (i) data from parent trial for all controls in the parent trial; (ii) data from parent trial for the subset of controls that also participated in this substudy; and (iii) data from substudy for controls in this substudy. Shaded bands represent 95% credible intervals around each prevalence estimate. Monsoon and dry seasons are indicated with vertical dashed lines.
(PDF)

**S3 Fig. Comparison of the prevalence of acute respiratory infection among children under 5 years by month of follow-up in the control arms of the parent trial and this substudy.** Prevalence estimates shown separately for (i) data from parent trial for all controls in the parent trial; (ii) data from parent trial for the subset of controls that also participated in this substudy; and (iii) data from substudy for controls in this substudy. Shaded bands represent 95% credible intervals around each prevalence estimate. Monsoon and dry seasons are indicated with vertical dashed lines.
(PDF)

**S4 Fig. Comparison of the prevalence of bruising/abrasion among children under 5 years by month of follow-up in the control arms of the parent trial and this substudy.** Prevalence estimates shown separately for (i) data from parent trial for all controls in the parent trial, (ii) data from parent trial for the subset of controls that also participated in this substudy; and (iii) data from substudy for controls in this substudy. Shaded bands represent 95% credible intervals around each prevalence estimate. Monsoon and dry seasons are indicated with vertical dashed lines.
(PDF)

## Acknowledgments

We would like to thank our dedicated field staff for their effort and the study participants for their generosity and time.

## Author Contributions

**Conceptualization:** Jesse D. Contreras, Benjamin F. Arnold, Jade Benjamin-Chung, Alan E. Hubbard, Mahbubur Rahman, Leanne Unicomb, Stephen P. Luby, John M. Colford, Jr, Ayse Ercumen.

**Data curation:** Jesse D. Contreras, Andrew Mertens.

**Formal analysis:** Jesse D. Contreras, Ayse Ercumen.

**Funding acquisition:** Benjamin F. Arnold, Stephen P. Luby, John M. Colford, Jr, Ayse Ercumen.

**Investigation:** Mahfuza Islam, Mahbubur Rahman, Ayse Ercumen.

**Methodology:** Jesse D. Contreras, Benjamin F. Arnold, Ayse Ercumen.

**Project administration:** Mahfuza Islam, Mahbubur Rahman, Ayse Ercumen.

**Resources:** Ayse Ercumen.

**Software:** Jesse D. Contreras, Benjamin F. Arnold, Jade Benjamin-Chung, Ayse Ercumen.

**Supervision:** Mahfuza Islam, Mahbubur Rahman, Ayse Ercumen.

**Validation:** Ayse Ercumen.

**Visualization:** Jesse D. Contreras.

**Writing – original draft:** Jesse D. Contreras.

**Writing – review & editing:** Jesse D. Contreras, Mahfuza Islam, Andrew Mertens, Amy J. Pickering, Benjamin F. Arnold, Jade Benjamin-Chung, Alan E. Hubbard, Mahbubur Rahman, Leanne Unicomb, Stephen P. Luby, John M. Colford, Jr, Ayse Ercumen.

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
