## [Editor Report · Decision Letter 0]

20 Jan 2022

Dear Dr Ercumen, 

Thank you for submitting your manuscript entitled "Longitudinal effects of an on-site sanitation intervention on child diarrhea and acute respiratory infection 1 to 3.5 years after implementation in a cluster-randomized controlled trial in rural Bangladesh" for consideration by PLOS Medicine.

Your manuscript has now been evaluated by the PLOS Medicine editorial staff and I am writing to let you know that we would like to send your submission out for external assessment.

However, before we can send your manuscript for assessment, we need you to complete your submission by providing the metadata that is required for full assessment. To this end, please login to Editorial Manager where you will find the paper in the 'Submissions Needing Revisions' folder on your homepage. Please click 'Revise Submission' from the Action Links and complete all additional questions in the submission questionnaire.

Please re-submit your manuscript within two working days, i.e. by Jan 24 2022 11:59PM.

Once your full submission is complete, your paper will undergo a series of checks in preparation for assessment. 

Kind regards,

Richard Turner, PhD

rturner@plos.org

---

## [Decision Letter · Decision Letter 1]

17 Mar 2022

Dear Dr. Ercumen,

Thank you very much for submitting your manuscript "Longitudinal effects of an on-site sanitation intervention on child diarrhea and acute respiratory infection 1 to 3.5 years after implementation in a cluster-randomized controlled trial in rural Bangladesh" (PMEDICINE-D-22-00199R1) for consideration at PLOS Medicine. 

Your paper was discussed with an academic editor with relevant expertise and sent to independent reviewers, including a statistical reviewer. The reviews are appended at the bottom of this email and any accompanying reviewer attachments can be seen via the link below:

[LINK]

In light of these reviews, we will not be able to accept the manuscript for publication in the journal in its current form, but we would like to invite you to submit a revised version that addresses the reviewers' and editors' comments fully. You will appreciate that we cannot make a decision about publication until we have seen the revised manuscript and your response, and we expect to seek re-review by one or more of the reviewers. 

We hope to receive your revised manuscript by Apr 07 2022 11:59PM. Please email us (plosmedicine@plos.org) if you have any questions or concerns.

Please let me know if you have any questions, and we look forward to receiving your revised manuscript. 

Sincerely,

Richard Turner PhD

Senior editor, PLOS Medicine

rturner@plos.org

Please adapt the title to better reflect journal style, and we suggest: "Evaluation of an on-site sanitation intervention against child diarrhea and acute respiratory infection outcomes 1 to 3.5 years after implementation in rural Bangladesh: Extended follow-up of a cluster-randomized controlled trial".

You use the descriptor "substudy" in the text, and as an alternative this could be included in the title for consistency.

In the abstract and throughout the paper, please quote p values alongside 95% CI, where available. 

Please add a new final sentence to the "Methods and findings" subsection of the abstract, which should begin "Study limitations include ..." or similar and should quote 2-3 of the study's main limitations. 

At line 60 (abstract) and at other points in the paper, please soften the language to reflect the research design and the observed changes. We suggest amending the text to "The apparent benefit of the WASH ...". 

After the abstract, please add a new and accessible "Author summary" section in non-identical prose. You may find it helpful to consult one or two recent research papers in PLOS Medicine to get an impression of the preferred style. 

Thank you for providing information about your analysis plan. Please highlight any analyses that were not prespecified. 

At line 130, please adapt the text to "We aimed to ..." or "We assessed ...".

In the Methods section, it may be worthwhile subdividing the subsection on "Statistical analysis" into two subsections on "Study procedures" and "Statistical analysis" or similar. 

Please remove the "Role of the funding source" statement at the end of the Methods section. 

Again at line 295, please temper the language and quote, alongside the apparent reduction of 19%, the point estimate and 95% CI.

At line 295 and elsewhere, please use the style "... under 5 years".

Please adapt reference call-outs to the following style throughout the paper: "... in children [1,2]." (noting the absence of spaces within the square brackets). 

Please remove the information on funding, competing interests and data sharing from the end of the main text. In the event of publication, this information will appear in the article metadata, via entries in the submission form. 

In the reference list, please convert all italics and boldface to plain text. Where appropriate, please list 6 author names rather than 3, followed by "et al.".

Is reference 27 a paper in preparation? If so, please supply it with your revision for the reviewers to take into account, or make it available as a preprint. 

Thank you for providing a completed CONSORT checklist. Please adapt this so that individual items are referred to by section (e.g., "Methods") and paragraph numbers, not by page or line numbers as these generally change in the event of publication. 

Please break the checklist out into an individual file, labelled "S1_CONSORT_Checklist" or similar and referred to as such in the Methods section (main text). 

Comments from the reviewers:

*** Reviewer #1:

Statistical review

This paper reports a long-term follow-up of a substudy from a cluster randomised trial investigating a sanitation intervention.

Generally the study was written and reported well. I have some minor comments:

1. Abstract: "There were no differences in intervention effects between periods with high-intensity vs. low-intensity/no promotion." - I would add 'significant' here to indicate this was formally tested.

2. Substudy Design: Was there a statistical justification for the substudy sample size? I would add a brief note on the justification whether it was statistical or based on feasibility (I note later that the authors show the effect size the substudy has 80% power for).

3. Statistical analysis: can the authors comment on missing data and how this was accounted for (presumably the analysis model is valid under a MAR assumption).

4. Although not strictly significant, the negative control item appears marginally significant from the CI - is this concerning to the authors about the interpretation of the primary outcome? This is briefly remarked upon in the discussion so may need nothing additional adding.

5. Line 283-284: there is evidence the effect is modified by season, but the interaction pvalue for index household status isn't significant: so I'm not sure it's correct to say there was evidence of effect modification for index household status. Perhaps the authors could make clear the evidence for this is stronger for season than index household?

James Wason

*** Reviewer #2: 

The authors present an interesting sub-study of the WASH-B Bangladesh trial that aims to estimate the effect of the sanitation arm of the intervention over a longer time period (3.5 years) than the original trial. The authors rightly point out that this work has the potential to help fill a current gap in the literature: we know very little about the long-term sustainability of WASH intervention effects. However, the main analysis pools results from 1-3.5 years post intervention, a period that overlaps with the parent trial, leaving the reader to ponder whether the intervention effects reported here truly reflect long-term intervention effects or whether they are skewed by measurements taken 1-2 years post-intervention (a period during which an intervention effect has previously been reported). Further specific comments follow. 

Abstract - 

The prevalence estimates - clarify that these are pooled estimates. 

Methods

Intervention 

- If possible, it would be useful to include an approximate number of visits/month during years - 2-3 when visits begin to taper off. 

- Lines 166- 168: How long after the intervention was implemented was use/access of latrines and use of saniscoops/potties measured? 

Statistical analysis

- No adjustment for other covars (like child age) given good balance at baseline - but there would have been births into compounds/household and some attrition over the 2.5 year follow-up. How were changes in the population handled and could this bias results? 

Results

- Lines 246 - 249: You switch between discussing enrolled households and descriptive statistics for compounds. It might be useful to clarify in the methods that there was one index household per compound (if this was the case). Also, it might be useful to provide some sort of population normalised measure of number of latrines per compound (latrines/person or latrines/household) to better understanding sharing. 

- Lines 250-252: Did you measure upgrades to latrines in control households over the same period? 

- Lines 257 - 262: Was there any differential movement in or out of the study population? i.e. losses to follow-up or differences in births/movement into study sites. 

- Line 267: It would be useful to mention the follow-up rounds during which bruising/abrasion was highest and lowest as is done for diarrhea and ARI.

- Intervention effects section: As the introduction speaks at length about the gap in the literature around measuring intervention effects over extended periods of times, it would seem relevant to include results that speak more directly to this gap, i.e. intervention effects over time/ at individual time points that extend past the original trial results. This will reduce sample size substantially but may still be informative. 

- Line 283: Would clarify what "index house status" means as I had to go back through methods. Also, as intervention promotion was targeted at the index household, is it possible that promotion intensity would be more important in this targeted subgroup of the study population? 

Discussion: 

- Lines 306 - 309: Could the difference between index and non-index households be due to differences in the physical infrastructure provided to index vs non-index households rather than the difference in promotion? Especially since you see no effect modification by promotion intensity. Additionally, in lines 313-324 you suggest that it is infrastructure improvements that may be the reason for the reduction in diarrhea. 

- Lines 314 - 315: In the results section you state that 97% control compounds and 100% intervention compounds had a latrine. Is this referring to the same statistic? If so, please clarify if you mean compound or household.

- Lines 314 - 319: The onset of the substudy occurred 1 year post-intervention. This means intervention sites would have been subject to 1 year of "intensive promotion" prior to collection of the data referenced in lines 314 - 316? It would be more informative to include latrine use habit prior to intervention implementation. 

- Line 319: The "backdrop" described is 1-year post-intervention as far I understood from the study design, so this sentence is confusing. To make this point, it would be useful to include data from the baseline phase of the parent study if available. 

- Lines 321 - 323: Clarify the unit of measurement - is this all existing latrines in a compound? Just latrines for index households? 

- Lines 323 - 324: Again, I'm not sure this follows if the reference point for "a setting with existing latrine access and latrine use" is 1 year post-intervention (see first paragraph of results section where it is explained that the data are taken 1-year post-intervention). 

- Lines 335: Is the age distribution between the two groups being compared similar (parent trial vs sub-study)? If not, this could be a potential reason for the discrepancy?

- Given the difference in promotional activities & diarrhea effects between the Kenya and Bangladesh sites, it seems odd that it is not referenced in the discussion of the role of intervention promotion. 

Figure S1: Rather than follow-up #, include indication of months post intervention. Follow-up round number is not informative in this figure. 

*** Reviewer #3: 

General Comments

This manuscript presents a robust analysis of the longitudinal effects of a sanitation intervention on child diarrhea and respiratory infections. The finding of a persistent reduction in diarrhea prevalence more than three years after the initial delivery of the intervention is well supported and provides confirmatory evidence for the shorter-term effect reported previously from the parent study. A truly interesting, and unexpected, observation arising from this sub-study is the substantially and consistently higher baseline risk for all caregiver-reported outcomes (including the negative control) compared with the parent study, which the authors explore in commendable depth. The description of these two parallel, partially overlapping assessments of the same outcomes using similar instruments is a valuable contribution in its own right, illustrating important discrepancies and raising critical questions relevant to the design and analysis of complex environmental health interventions in real world settings.

Specific Comments

L69: Missing a verb in "when latrine use practices in place"

L204: Please state how prevalence estimates and CIs were computed; the captions to Figures 1, 2, and S3-S5 mention "credible intervals", which would suggest some sort of Bayesian approach was used.

L228 - 232: Was any effort made to differentiate between promotion intensity and child age? As I understand the study design, the low/no promotion intensity period would coincide with children being a year or two older, on average, than they were during the high promotion intensity period.

Table 2: It is helpful to see EMM assessed on both multiplicative and additive scales—well done! However, why is the dry-season diarrhea PD positive when the raw prevalence estimates and the PR all indicate lower prevalence in the sanitation arm during the dry season?

L338-340: Were the parent study and sub-study data collection visits ever performed concurrently or within the same 7-day period?

L350-351: This interpretation is compelling. However, can any arguments be made in favor of the parent study providing more accurate measurement of these caregiver-reported outcomes?

Figure S2: Are the observation counts for the parent study across all children (including those not enrolled in this sub-study), or among sub-study children only?

***

[LINK]

---

## [Decision Letter · Decision Letter 2]

25 May 2022

Dear Dr. Ercumen,

Thank you very much for re-submitting your manuscript "Evaluation of an on-site sanitation intervention against childhood diarrhea and acute respiratory infection 1 to 3.5 years after implementation: Extended follow-up of a cluster randomized controlled trial in rural Bangladesh" (PMEDICINE-D-22-00199R2) for consideration at PLOS Medicine.

I have discussed the paper with our academic editor and it was also seen again by three reviewers. I am pleased to tell you that, provided the remaining editorial and production issues are fully dealt with, we expect to be able to accept the paper for publication in the journal.

[LINK]

Please let me know if you have any questions, and we look forward to receiving the revised manuscript.   

Sincerely,

Richard Turner, PhD

rturner@plos.org

Requests from Editors:

Please finalize the arrangements for data deposition. 

At line 38, we suggest "... composite sanitation intervention" or similar. 

Again at line 38, we ask you to soften the wording slightly, to "... led to a reduction of diarrhea by 39% and of ARI by ...", say. Please make similar changes around line 141.

At line 52 and elsewhere, please use the style "... 1 year of ..." although numbers should be spelt out at the start of sentences. 

At line 60, is there a way to avoid "0.000"?

At line 68, we ask you to adapt the text to "... appeared to be sustained ..." or similar, considering the statistical evidence seen. 

At line 90 (author summary) and any other instances, please make that "... under 5 years". 

At line 159, please make that "... we used". 

Should the numbers at line 296-7 not add up to 9,088?

At line 349, please remove "by 19%" (which is implicit in the quoted PR). 

Given the observed p value, please also soften the wording here to "... was associated with a reduction in the prevalence of ...".

Throughout the text, please ensure that reference call-outs precede punctuation (e.g., "... among children [4].").

Please finalize the citation for reference 27. Prior to publication of the present paper, this reference will either need to be available in published form, made available as a preprint, or removed. 

Is reference 29 missing a journal or source name?

Please rename the attached CONSORT checklist "S1_CONSORT_Checklist".

Thank you for including the Islam et al draft paper, and this can now be removed. 

Comments from Reviewers:

*** Reviewer #1: 

Thank you to the authors for addressing all my previous comments well. I have no further issues to raise.

*** Reviewer #2: 

I am satisfied with the author responses to my comments and thank the authors for their thoughtful and comprehensive responses. I enjoyed reading and reviewing the paper. 

*** Reviewer #3: 

The authors have satisfactorily addressed the previous reviewer comments.

***

[LINK]

---

## [Editor Report · Decision Letter 3]

2 Jun 2022

Dear Dr Ercumen, 

On behalf of my colleagues and the Academic Editor, Dr Bhutta, I am pleased to inform you that we have agreed to publish your manuscript "Evaluation of an on-site sanitation intervention against childhood diarrhea and acute respiratory infection 1 to 3.5 years after implementation: Extended follow-up of a cluster randomized controlled trial in rural Bangladesh" (PMEDICINE-D-22-00199R3) in PLOS Medicine.

Prior to final acceptance, please substitute the new label for the attachment referred to at line 214.

PRESS

Sincerely, 

Richard Turner, PhD 

rturner@plos.org